# Change in Asthma Is Associated with Change in PTSD in World Trade Center Health Registrants, 2011 to 2016

**DOI:** 10.3390/ijerph19137795

**Published:** 2022-06-25

**Authors:** Stephen M. Friedman, Howard Alper, Rafael E. de la Hoz, Sukhminder Osahan, Mark R. Farfel, James Cone

**Affiliations:** 1World Trade Center Health Registry, New York City Department of Health and Mental Hygiene, Long Island City, New York, NY 11101, USA; halper@health.nyc.gov (H.A.); osukhmin@health.nyc.gov (S.O.); mfarfel@health.nyc.gov (M.R.F.); jcone@health.nyc.gov (J.C.); 2Division of Occupational Medicine, Icahn School of Medicine at Mount Sinai, New York, NY 10029, USA; rafael.delahoz@mssm.edu

**Keywords:** World Trade Center Health Registry, disaster, asthma, ACT, PTSD, quality of life

## Abstract

The WTC Health Registry (WTCHR) is a closed, longitudinal cohort of rescue/recovery workers and survivors exposed to the 11 September 2001 disaster. WTCHR enrollees diagnosed with asthma after 11 September 2001 continued to experience poor control despite treatment. Asthma is associated with mental problems, although their bidirectional movement has not been studied. This study tested whether a clinical change in mental problems was associated with a difference in asthma control, and whether a change in asthma control varied with a change in quality of life (QoL). Difference in the Asthma Control Test (ACT) on the WTCHR from 2011–12 to 2015–16 was compared with the change in the Post-traumatic Stress Disorder Checklist (PCL-17), the Patient Health Questionnaire depression scale, self-reported heartburn, and change of physical and mental QoL over this period. In adjusted multinomial multivariable logistic regression, improved PCL-17 was associated with a better ACT score, odds ratio (OR) = 1.42 (95% C.I. 1.01, 1.99), and a worsened PCL-17 score was associated with a worsened ACT score, OR = 1.77 (95% C.I. 1.26, 2.50). Decreased ACT was associated with poor physical QoL, OR = 1.97 (95% C.I. 1.48, 2.62). Change in mental health measures tracked with change in asthma control, which correlated with a change in QoL. Careful follow-up and treatment of all three are indicated to improve these inter-related issues.

## 1. Introduction

New incident asthma is one of the hallmarks of World Trade Center (WTC)-disaster-related illness. Symptoms of asthma, as well as bronchial hyperreactivity and decreased lung function were reported early on in Fire Department of New York (FDNY) firefighters, especially those exposed on the day of the disaster [1,2]. Increased asthma symptoms were noted in exposed Manhattan residents 5–9 weeks after the attack [3]. Wheeler et al. reported a significant increase in newly diagnosed self-reported asthma among rescue/recovery workers with earlier arrival at the site and longer duration of exposure to the dust cloud and work on the debris pile [4]. Brackbill et al. found that the increased rate of newly diagnosed asthma was also seen in lower Manhattan residents and area workers who were exposed to the dust cloud, experienced a heavy layer of dust at home or office, or remained at home after the disaster [5].

Asthma has continued to be reported among WTC-exposed groups. As of 2009, 8.8% of exposed firefighters had FDNY physician-diagnosed asthma [6]. Apart from these firefighters, among 7027 rescue/recovery workers and volunteers followed for nine years, the cumulative incidence of asthma was 27.6% [7].

Asthma is known to be associated with mental health disorders [8,9,10]. ln a longitudinal study of WTC rescue/recovery workers, probable post-traumatic stress disorder (PTSD), as measured by the PTSD Checklist-17 (PCL-17) at baseline, was associated with self-reported physician-diagnosed new onset asthma over a three to five year follow-up period, and that the association was much stronger for the more chronic PTSD trajectories within that interval [11].

Studies have indicated that asthma control in WTC-disaster-exposed populations may be poor, as measured by National Asthma Education and Prevention Program Guidelines [12]. Among 2445 WTCHR survey participants examined in 2011–12, Jordan et al. found that 33.7% had poorly controlled symptoms and 34.6% had very poorly controlled symptoms [13]. A higher number of mental health conditions (PTSD, depression, and anxiety), gastroesophageal reflux symptoms (GERS), and obstructive sleep apnea (OSA), as well as increased age, lower educational attainment, current smoking, participation in rescue/recovery work, and obesity were all significantly associated with very poor asthma control. In a study of 218 WTC rescue and recovery workers, Xu et al. found that higher gastroesophageal reflux disease (GERD) scores and PTSD, as well as increased age, lower income, and high WTC exposure were associated with very poor asthma control [14]. The highly prevalent comorbidities reported in WTC workers [15] are known to be associated with difficult-to-treat asthma in the general population [16]. Wyka et al. demonstrated that lower respiratory symptoms and PTSD in enrollees in the WTCHR exhibited a bidirectionality in these physical and mental health problems [17]. Psychological variables have also been associated with asthma-related quality of life [18,19]. However, the effect of change in level of asthma control on health-related quality of life has not yet been reported in WTC-exposed individuals. These findings raised the new question of whether a change in PTSD, depression, and quality of life over time would be accompanied by a concomitant change in the level of asthma control.

The primary hypothesis of this study was that a clinically significant change from the W3 survey to the W4 asthma survey in risk factors would be associated with a clinically significant change in asthma control level. A second hypothesis was that clinically significant change in asthma control would be associated with clinically significant change in quality of life measures.

## 2. Materials and Methods

### 2.1. Data Source

The WTC Health Registry (WTCHR) is a closed, longitudinal cohort of rescue/recovery workers, volunteers, local area workers, residents, students, teachers, and passersby who met criteria for exposure to the disaster on or after 11 September 2001 [20]. A total of 68,046 adults aged ≥18 on 11 September 2001 responded to the first registry survey (Wave 1 or W1, 2003–2004). To date, four surveys have been completed, with 46,019 enrollees responding to the second survey (Wave 2, 2006–2007), 42,395 responding to the third survey (Wave 3, 2011–2012), and 35,958 responding to the fourth survey (Wave 4, 2015–2016). These surveys asked about demographic characteristics, exposure to the disaster, physical and mental health status, self-report of diagnosed illnesses, social status, and quality of life. The W3 survey included an asthma module that inquired about symptoms, diagnosis date and control level using a modified version of the Asthma Control Test (ACT) [21,22].

### 2.2. Analytic Sample

There were 10,583 enrollees who reported that they had been diagnosed with asthma between 11 September 2001 through 2012 on the Wave 1, 2, or 3 surveys. Those who completed the asthma symptoms and control module in Wave 3 (*n* = 5379) were mailed the Wave 4 asthma survey. Of these, 69.4% (3735) answered the W4 asthma survey. There were 2796 respondents who completed the components of the ACT survey in the Wave 3 and Wave 4 asthma surveys, and they comprise the cohort for this study (Figure 1). Wave 3 characteristics of the participants and non-participants in the W4 asthma survey were compared using the chi-square test.

The asthma surveys asked about asthma morbidity and symptom control. The difference between the modified ACT used in Wave 3, 2011–12, and the ACT in the Wave 4 asthma survey in 2016 was that the modified ACT asked about symptoms in the last 30 days, whereas the ACT asked about symptoms in the last 28 days. All self-reported data reported in this study were acquired through these surveys.

The Institutional Review Boards of the New York City Department of Health and Mental Hygiene and the Centers for Disease Control and Prevention approved the Registry protocols.

### 2.3. Change in Asthma Control

Change was measured by characterizing whether the change in ACT score was clinically significant, i.e., a change of at least three points versus within three points up or down [22], or by whether the change in ACT resulted in a change in the level of control (very poorly controlled, 0–15; poorly controlled, 16–19; or controlled, 20–25). 

### 2.4. Predictors of Change in Asthma Control

Demographic variables including age on 11 September 2001, sex, and race/ethnicity were recorded in W1. Race/ethnicity was categorized in two different ways: (1) five groups: non-Hispanic White, non-Hispanic African American, non-Hispanic Asian, and other, and (2) two groups: non-Hispanic White versus others. 

Since the goal was to measure the association between clinically significant change from W3 to W4 in asthma symptom control risk factors and clinically significant change in asthma control level, we first defined significant change in the predictors. Seven factors potentially associated with control level were measured at both W3 and W4 (see Table 1). The effect of these factors on control level was examined using the change in the factor from W3 to W4 as the independent variable. These time dependent variables included the time interval from W3 to W4, score on the 17-item PTSD Checklist, Stressor-Specific Version (PCL-17) [23,24], score on the 8-item Patient Health Questionnaire depression scale (PHQ-8) [25], self-reported body mass index (BMI), gastroesophageal reflux symptom (GERS) frequency in the past 12 months, and level of social support. (The PCL-17 is scored as 17 to 85 with ≥44 as positive; the PHQ-8 is scored as 0 to 24 with ≥10 as positive). Difference in BMI was categorized as relative change of at least 5 percent. For GERS in the last 12 months, change was categorized as crossing a threshold of symptoms of at least twice a week. Social support was estimated based on four criteria: having ≥3 close friends, having contact with those friends ≥ twice a month, attending religious service ≥ twice a month, and being actively involved in a group or club. Social support was scored as low (meeting none or 1 of the criteria), medium (meeting 2 of the criteria) or high (meeting ≥3 of the criteria). Change was measured as a minimum of movement from one level to the next up or down (e.g., from low to medium or low to high). For PTSD, a change of at least 10 points in the PCL-17 score was required [25,26,27] as the standard deviation for the difference was 10.44. For depression using the PHQ-8, a difference of at least 5 points was used as the change criterion [25,28]. 

There was little change in the percentage of participants who were currently smoking from W3 to W4. Only 34/157 (21.7%) adults who smoked at W3 stopped at W4, and only 17/2918 (0.58%) adults who did not smoke at W3 reported smoking every day at W4Smoking status at W3, rather than change in smoking status was used to provide the effect of smoking on asthma control.

Questions about healthcare accessibility were revised from W3 to W4 so change in healthcare access could not be determined, and access at W3 was used. In W3, poor healthcare accessibility was defined as the absence of health insurance, absence of a personal health care provider, or not receiving needed physical health care in the past 12 months. 

OSA prevalence was based on self-report of a history of physician diagnosis which should not change negatively from W3 to W4 (once a diagnosis is recorded in W3, it is unlikely to be undone in W4), so OSA at W3 was used.

### 2.5. Statistical Analysis

The bivariate association between possible risk factors and change in ACT score from W3 to W4 was measured using the chi-square test. The association between possible risk factors and change in ACT score was also modeled using bivariate or multivariable logistic regression analyses predicting clinically significant change in ACT score from W3 to W4. The ACT change reference category was chosen to be ±2, and the probability of being assigned to the ≤−3 or ≥3 categories were compared with this reference, using odds ratios and their 95% confidence intervals. A proportional odds model was attempted, but the data did not support the proportional odds assumption. Subsequently, a multivariable multinomial logistic regression model was used. The output yielded four odds ratios for each pair of independent and dependent variables. 

A PCL-17 score change of ≥10 points and PHQ-8 score change of ≥5 points were included in separate models since these variables are collinear.

The relationship of ACT score changes and quality of life measures was analyzed using a multivariable multinomial logistic regression model where a clinically significant change in ACT score was used to predict a change of ≥10 days of poor physical or mental health (versus within 9 days up or down). A sensitivity analysis was undertaken using a cutoff of 14 or more days of poor physical or mental health as the dependent variable [29].

### 2.6. Participants vs. Non-Participants in W4

Between participants and non-participants in W4, the prevalence of controlled versus not controlled (poorly or very poorly controlled) asthma at W3 measured by ACT criteria was not significantly different (52.1% in participants versus 50.7% in non-participants) (*p* = 0.3976). However, there were some differences in covariates. Participants were more likely to report being older (*p* < 0.0001), White non-Hispanic race and ethnicity (*p* < 0.0001), and higher educational attainment (*p* < 0.0001). Participants were less likely to report current smoking at W3 (*p* < 0.0001), less likely to have poor access to healthcare (*p* = 0.0126), lower social support (*p* = 0.0003), or a history of less GERS (*p* = 0.0065). Participants and non-participants did not vary significantly by sex, obesity status, rescue/recovery worker experience, dust cloud exposure after 11 September 2001, or reported history of OSA.

## 3. Results

The mean age at submission of the W3 survey was 52.0 years, standard deviation (SD) 9.67 years, and for the W4 asthma survey, 56.0 years, SD 9.66 years. The mean interval from the Wave 3 to the Wave 4 asthma interview was 4.11 years, SD 0.20 years. The mean time interval between interviews was not associated with change in ACT score from the W3 survey to the W4 asthma survey (Pr > F = 0.081).

Asthma control decreased from W3 to the W4 asthma survey. The mean change in ACT score was −1.07, range −18 to +18; 32.6% scored higher, 16.2% scored the same, and 51.2% scored lower. Using a clinically significant change of three points, 15.9% scored higher, 52.5% scored the same, and 31.6% scored lower. Mean BMI increased by 0.19 kg/m^2^ (95% CI 0.082, 0.30), mean PCL-17 score decreased by 2.0 points (95% CI 1.55, 2.41), and mean PHQ-8 decreased by 0.32 points (95% CI 0.13, 50).

The results of the bivariate analyses of demographics, covariates and change of asthma control between the W3 and W4 asthma surveys are presented in Table 2 and Table 3. Associations related to factors measured at a single time point (i.e., W1 or W3) are shown in Table 2. Age group on 9/11, sex, education, and 9/11 exposure category documented at Wave 1 were associated with a clinically significant change in ACT score (up or down by ≥3 points) from W3 to W4. Race/ethnicity was not significantly associated with a change in ACT score when it was categorized into five groups. However, there was significant variability when using the non-Hispanic White versus the remaining groups, so this breakdown was presented in the tables. Other factors measured at W3, income level, poor access to healthcare, GERS frequency, obstructive sleep apnea, PCL-17 ≥ 44, and PHQ-8 ≥ 10, were also associated with a change in ACT score. ACT level at W3 was strongly inversely related to a change in ACT score, i.e., those with poorly controlled asthma at W3 were more likely to have improvement in ACT score from W3 to W4. Similarly, those with controlled asthma at W3 were less likely to improve their ACT score in W4. The year of asthma diagnosis and dust cloud exposure, smoking status, BMI category, and level of social support at W3 were not associated with a change in ACT score.

Possible risk factors measured as change from W3 to W4 are presented in Table 3. Change in PCL-17 score by ≥10 points, change in PHQ-8 score by ≥5 points, and change in GERS frequency in the past 12 months across the twice-a-week threshold were significantly associated with a change in ACT scores from W3 to W4. Change in income level, smoking history, BMI, and social support category were not associated with a change in ACT score.

By bivariate logistic regression (Table 4), female sex, very poorly controlled asthma at W3, improving GERS, improving PCL-17 score, and improving PHQ-8 score were associated with an improved ACT score from W3 to W4. Older age group on 9/11, history of sleep apnea at W3, poorly controlled asthma at W3, and worse GERS, PCL-17, and PHQ-8 scores from W3 to W4 were associated with a worsening ACT score. Race/ethnicity, smoking history, dust cloud exposure, BMI change of ≥5%, and social support change were not associated with a change in ACT score. Not being a rescue/recovery worker was protective against worsening asthma control. Poor healthcare accessibility at W3 was associated with an improved ACT score but not with worsened ACT.

In the adjusted multinomial multivariable logistic regression analysis (Table 4), as in the bivariate analyses, female sex and uncontrolled asthma at W3 were associated with an improved ACT score from W3 to W4. Improving PCL-17 was associated with an improving ACT score (adjusted odds ratio (ORadj) = 1.42 (95% C.I 1.01, 1.99), and worsening PCL-17 score was associated with a worsening ACT score (ORadj = 1.77 (95 % C.I. 1.26, 2.50). Older age group and increasing frequency of GERS from W3 to W4 were associated with a worsening ACT score, and 9/11 exposure group other than rescue/recovery worker and higher educational attainment were protective against a worsening ACT score.

In an adjusted logistic regression model where clinically a significant change in PHQ-8 was substituted for a clinically significant change in PCL-17, worsening PHQ-8 score approached but did not reach statistically significant association with worsening ACT score, ORadj = 1.34, 95%, confidence intervals (C.I. 0.96, 1.86).

The effect of a clinically significant change in ACT score on quality of life was examined in a multivariable multinomial logistic regression model (Table 5). Decrease (≥3 points) in ACT score was associated with a change of ≥10 days of poor physical health, ORadj = 1.97 (95% CI 1.48, 2.62). The possible association between improvement in ACT score (≥3 points) and fewer days of poor physical health approached but did not reach significance, ORadj = 1.38 (95% CI 0.96, 1.98). Change in ACT score was not associated with a change in days of poor mental health. When PCL-17 was removed from the model for mental health quality of life, a worse ACT score (≥3 points) was associated with worse mental health (≥10 days), ORadj = 1.44 (95% C.I. 1.07, 1.96).

## 4. Discussion

This study is the first in the WTC disaster literature to demonstrate bidirectionality in clinically significant change in PTSD and ACT scores over time. ACT score change paralleled change in PTSD score, depression score, and GERS frequency in bivariate and in multivariable logistic regression analyses accounting for covariates.

Previous studies have demonstrated an association between mental health conditions and incident asthma or poor asthma control [11,13,14]. The current study goes further to demonstrate that clinically meaningful change in these mental health indicators is associated with concomitant change in asthma symptom control. The indicators used in this study, 3-point change in ACT score, change to GERS to at least twice a week, 10-point change in PCL-17 score, and 5-point change in PHQ-8 score were meaningful by clinical standards and sensitive enough to identify change from W3 to W4.

The strongest predictor of change in ACT score was the ACT score at W3. Enrollees with very poorly controlled asthma at W3 were several times more likely than those with controlled asthma to show improvement in ACT score. This pattern might be expected since there is a greater possibility for improvement among those with very poor control than those with controlled asthma, and similarly, greater possibility for worsening control among those with controlled asthma than those with very poor control. Even so, the association between change in PTSD score and change in ACT score remained when baseline ACT score at W3 was accounted for in the adjusted model.

Poor healthcare access has been shown to be associated with poor asthma control [13]. In the current study it was paradoxically associated with improvement in the ACT score in the bivariate linear regression model, despite efforts by the WTCHR to link care to the World Trade Center Health Program. This finding was unexpected and requires further investigation.

Asthma is known to be associated with poor quality of life, in both community and WTC populations [14,30,31]. Similarly in this study, the ACT change tracked a clinically significant change in quality of life measures. A decrease of ≥3 points in the ACT was associated with a nearly twofold increased adjusted odds of poorer physical health. A change of ≥3 points in ACT score was associated with a change in days of poor mental health only when changes in PCL-17 and PHQ-8 scores were omitted from the model. A mediation study of the effect of asthma change on mental health quality of life through the association between asthma control and mental health might be revealing.

Previous WTCHR reports and studies of broader populations have also shown that report of poor mental health precedes report of lower respiratory symptoms and poor quality of life [9,10,32,33]. In this study data on asthma control, mental health covariates and quality of life were collected at approximately the same point in time, suggesting that these factors may be affecting each other.

### Limitations

A weakness of this study is that all the diagnoses and measures were self-reported. However, key measures (ACT score, PCL-17, PHQ-8, days of poor quality of life) are well-validated, and clinically significant thresholds have been determined [25,26,27,28,29,34,35]. Analyses using slightly different measures for change in PCL-17 (crossing the 44-point threshold for positive PTSD), change in ACT score (change in controlled, poorly controlled, and very poorly controlled categories), and change in physical and mental quality of life (crossing the 14-days-per-month threshold of poor physical or mental health), gave similar results. Nevertheless, measurement of asthma control level in this study would benefit from objective clinical measurement such as spirometry with bronchodilator response [36] or forced oscillometry [37]. Still, this study provided longitudinal, prospective data that allowed for measurement of the relationships among respiratory, mental health, and quality of life variables.

The participants who self-reported asthma in the W3 asthma module and the Wave 4 asthma survey were older, had completed a higher level of education, and reported fewer mental health problems than those who did not participate. These differences may limit generalizing the findings to the entire WTCHR population. Still, 69% of enrollees who completed the W3 asthma module did participate in this study and represented a substantial sample for this analysis.

The association between mental health problems and asthma has been well-reported both in those exposed to the WTC disaster and in the general population [9,10,11,13,14,36]. Possible mechanisms include changes in gene expression and stress modulation of the hypothalamic–pituitary–adrenal cortex axis, autonomic nervous system, lung function, and the immune system [8,11,38,39]. However, the actual pathways remain to be elucidated.

Change in asthma control and other mental illnesses may also be bidirectional, as in the case of asthma and panic disorder [40]. In this study, the inter-relationship of asthma and PTSD underscored the potential importance of treating mental health problems and asthma simultaneously to improve the outcomes of both. Similarly, for asthma control and quality of life, improvement in one is likely to be accompanied by in improvement in the other. This study did not directly assess the effect of treatment of PTSD or asthma on the other. Although access through the WTC Health Program made multidisciplinary treatment possible for many exposed individuals, future studies will be needed to test whether treatment of PTSD reduces the symptoms of asthma with an accompanying change in quality of life.

This study antedated the COVID-19 pandemic, but it may have an ongoing effect on WTCHR enrollees with PTSD and asthma, and their quality of life. Subsequent WTCHR surveys will provide an opportunity to investigate the relationship between these health issues and COVID-19 in detail.

## 5. Conclusions

This prospective longitudinal study measured the relationships among asthma, mental health, and quality of life. Change in mental health paralleled change in asthma control and change in asthma control correlated with change in quality of life. Careful study of simultaneous treatment of asthma and mental illness are needed to achieve reciprocal, inter-related benefits.

## Figures and Tables

**Figure 1 ijerph-19-07795-f001:**
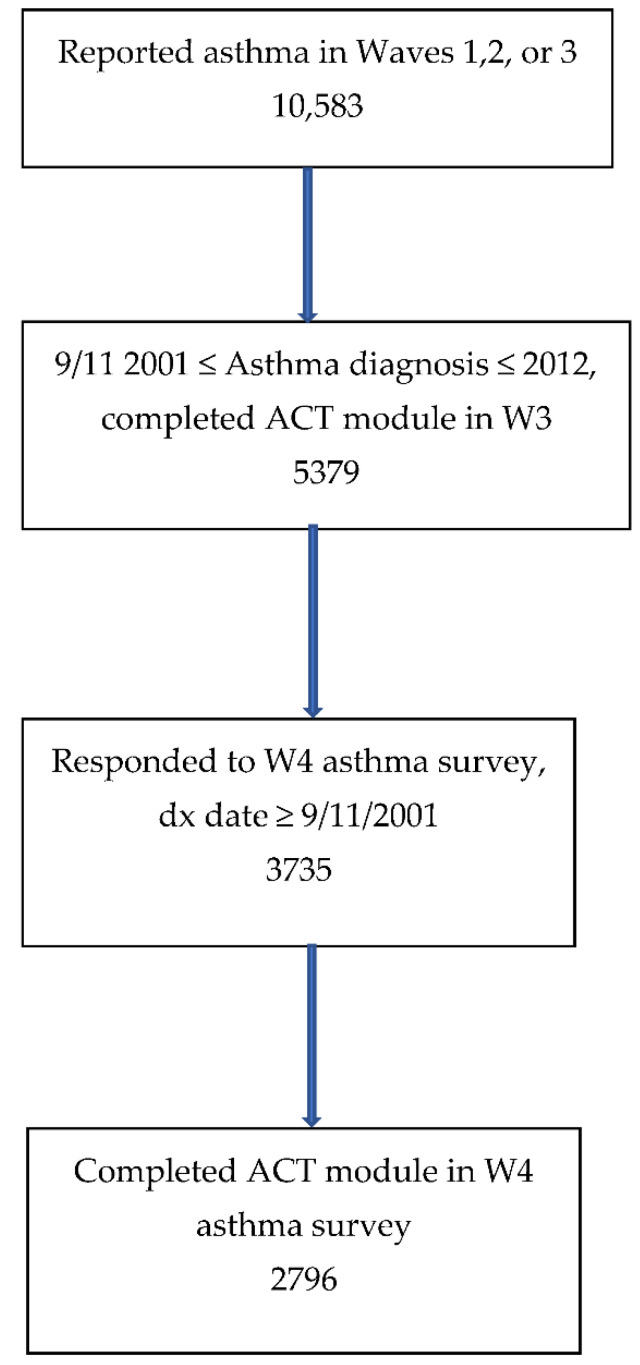
Flow diagram of participation in this study.

**Table 1 ijerph-19-07795-t001:** Change from Wave 3 to Wave 4.

Variable	Significant Change	Reference
ACT	Increase or decrease of 3 points or more	Remain within 2 points
Income level	Increase or decrease of one or more levels	No change
Smoking	Started smoking or quit smoking	No change
GERS frequency in last 12 months	Increase to two or more times a week or decrease to less than two times a week	Not crossing the ≥twice a week frequency threshold
BMI relative change	Increase or decrease of 5% or more	Remain within less than 5%
Social support	Increase or decrease of 2 levels or more	Remain within 1 level
PTSD-17	Increase or decrease of 10 points or more	Remain within 9 points
PHQ-8	Increase or decrease of 5 points or more	Remain within 4 points
Physical health QoL	Increase or decrease of 10 days or more	Remain within 9 days
	Increase or decrease of days to cross the 14 days/month threshold	Not crossing the 14 days/month threshold
Mental health QoL	Increase or decrease of 10 days or more	Remain within 9 days
	Increase or decrease of days to cross the 14 days/month threshold	Not crossing the 14 days/month threshold

**Table 2 ijerph-19-07795-t002:** Demographics, exposures and other relevant factors’ effect on change in ACT scores.

Measurement at Wave 1	Overall	Distribution by ACT Change from W3 to W4						Chi-Square *
		Decrease by ≥ 3		Within +/−3		Increase by ≥ 3		
	Number	Number	Percent	Number	Percent	Number	Percent	
**Age group 11 September**								
18–34	645	170	26.4	345	53.5	130	20.2	
35–39	508	171	33.7	259	51.0	78	15.4	
40–49	1026	341	33.2	538	52.4	147	14.3	
50-	617	202	32.7	326	52.8	89	14.4	
Total	2796	884	31.6	1468	52.5	444	15.9	***p* = 0.0071 ***
**Sex**								
Male	1650	557	33.8	875	53.0	218	13.2	
Female	1146	327	28.5	593	51.7	226	19.7	
Total	2796	884	31.6	1468	52.5	444	15.9	***p* < 0.0001**
**Race/ethnicity**								
White, non-Hispanic	1880	574	30.5	1019	54.2	287	15.3	
Black non-Hispanic, Hispanic, Asian, other	916	310	33.8	449	49.0	157	17.1	
Total	2796	884	31.6	1468	52.5	444	15.9	***p* = 0.0360**
**Education**								
High school or less	672	245	36.5	323	48.1	104	15.5	
Some college	822	287	34.9	394	47.9	141	17.2	
College and higher	1287	347	27.0	741	57.6	199	15.5	
Total	2781	879	31.6	1458	52.4	444	16.0	***p* < 0.0001**
**11 September exposure category**								
RR worker	1623	553	34.1	831	51.2	239	14.7	
Others	1173	331	28.2	637	0.0	205	17.5	
Total	2796	239	8.5	1468	52.5	444	15.9	***p* = 0.0026**
**Dust cloud exposure on 11 September**								
Yes	1502	535	35.6	855	56.9	277	18.4	
No	995	345	34.7	612	61.5	166	16.7	
Total	2497	880	35.2	1467	58.8	443	17.7	*p* = 0.2085
		**Distribution by ACT change**						
Measurement at W3	Overall	Decrease by ≥ 3		Within +/−3		Increase by ≥ 3		
**Income level**								
≤25,000	327	111	13.1	156	11.1	60	14.2	
25,000–50,000	434	145	17.1	226	16.1	63	14.9	
50,001–75,000	482	161	19.0	231	16.4	90	21.2	
75,001–150,000	1039	341	40.2	542	38.6	156	36.8	
>150,000	396	91	10.7	250	17.8	55	13.0	
	2678	849		1405		424		***p* = 0.0003**
**Smoking history**		Decrease by ≥ 3		Within +/−3		Increase by ≥ 3		
Currently smoking	258	94	36.4	118	45.7	46	17.8	
Previously but not currently smoking	972	296	30.5	527	54.2	149	15.3	
Never smoked	1378	432	31.3	731	53.0	215	15.6	
Total	2608	822	31.5	1376	52.8	410	15.7	*p* = 0.1989
**Poor health care accessibility**								
Yes	594	188	31.6	283	47.6	123	20.7	
No	2176	685	31.5	1172	53.9	319	14.7	
Total	2770	873	31.5	1455	52.5	442	16.0	***p* = 0.0009**
**Body mass index**								
Underweight/normal	591	166	28.1	331	56.0	94	15.9	
Overweight	1007	310	30.8	539	53.5	158	15.7	
Obese	1146	393	34.3	573	50.0	180	15.7	
Total	2744	869	31.7	1443	52.6	432	15.7	*p* = 0.0884
**GERS frequency**								
≤Once/week	2013	623	30.9	1093	54.3	297	14.8	
>Once/week	769	255	33.2	370	48.1	144	18.7	
Total	2782	878	31.6	1463	52.6	441	15.9	***p* = 0.0054**
**History of obstructive sleep apnea**								
Yes	805	283	35.2	395	49.1	127	15.8	
No	1856	558	30.1	1001	53.9	297	16.0	
Total	2661	841	31.6	1396	52.5	424	15.9	***p* = 0.0279**
**Social support level**								
Low	573	189	33.0	297	51.8	87	15.2	
Medium	1005	307	30.5	535	53.2	163	16.2	
High	1146	363	31.7	601	52.4	182	15.9	*p* = 0.8953
Total	2724	859	31.5	1433	52.6	432	15.9	
**PCL-17**								
≥44	901	305	33.9	418	46.4	182	20.2	
<44	1745	531	30.4	983	56.3	244	14.0	
Total	2646	836	31.6	1401	52.9	426	16.1	***p* < 0.0001**
**PHQ-8**								
≥10	819	268	32.7	389	47.5	162	19.8	***p* = 0.0001**
<10	1861	578	31.1	1022	54.9	261	14.0	
Total	2680	846	31.6	1411	52.6	423	15.8	
**ACT level at W3**								
Controlled	1440	485	33.7	862	59.9	93	6.5	
Poorly controlled	652	251	38.5	254	39.0	147	22.5	
Very poorly controlled	704	148	21.0	352	50.0	204	29.0	
	2796	884	31.6	1468	52.5	444	15.9	***p* < 0.0001**

* bolded text indicates *p* < 0.05.

**Table 3 ijerph-19-07795-t003:** Factors measured as the difference between W3 and W4 association between change in risk factor and change in ACT score.

Risk Factor Change, W3 to W4	Overall	Distribution of ACT Change from W3 to W4						Chi-Square *
		Decrease by ≥ 3		Within +/− 3		Increase by ≥3		
	Number	Number	Percent	Number	Percent	Number	Percent	
**Income level**								
no change	1383	443	32.0	729	52.7	211	15.3	
Decreased	320	114	35.6	165	51.6	41	12.8	
Increased	604	187	31.0	316	52.3	101	16.7	
Total	2307	744	32.2	1210	52.4	353	15.3	*p* = 0.4593
**Smoking**								
no change	2047	646	31.6	1094	53.4	307	15.0	
started smoking	32	12	37.5	16	50.0	4	12.5	
stopped smoking	66	23	34.8	30	45.5	13	19.7	
Total	2145	681	31.7	1140	53.1	324	15.1	*p* = 0.6508
**BMI**								
5% < change < −5%	1406	434	30.9	758	53.9	214	15.2	
increased by ≥ 5%	591	206	34.9	299	50.6	86	14.6	
decreased by ≥ 5%	422	124	29.4	225	53.3	73	17.3	
Total	2419	764	31.6	1282	53.0	373	15.4	*p* = 0.2827
**GERS frequency**								
remained the same	1950	597	30.6	1058	54.3	295	15.1	
increased to ≥ twice a week	323	134	41.5	147	45.5	42	13.0	
decreased to < twice a week	198	52	26.3	101	51.0	45	22.7	
Total	2471	783	31.7	1306	52.9	382	15.5	***p* < 0.0001**
**Social support level**							0.0	
within 2 levels	2112	665	31.5	1125	53.3	322	15.2	
worsened by ≥ 2 levels	102	39	38.2	42	41.2	21	20.6	
improved by ≥ 2 levels	125	36	28.8	68	54.4	21	16.8	
Total	2356	740	31.6	1235	52.8	364	15.6	*p* = 0.1706
**PCL-17 score**								
within 10	1600	492	30.8	880	55.0	228	14.3	
worsened by ≥ 10	426	120	28.2	201	47.2	105	24.6	
improved by ≥ 10	230	103	44.8	102	44.3	25	10.9	
Total	2256	715	31.7	1183	52.4	358	15.9	***p* < 0.0001**
**PHQ-8 score**								
within 10	1693	532	31.4	912	53.9	249	14.7	
worsened by ≥ 5	342	92	26.9	171	50.0	79	23.1	
improved by ≥ 5	253	105	41.5	118	46.6	30	11.9	
Total	2288	729	31.9	1201	52.5	358	15.6	***p* < 0.0001**

* bolded text indicates *p* < 0.05.

**Table 4 ijerph-19-07795-t004:** Bivariate and adjusted multivariable multinomial logistic regression for clinically significant change in ACT score versus demographics and clinically significant change in physical, and mental health factors from W3 to W4, odds ratios and 95% confidence intervals.

Multivariable Model: Change in ACT Score of ≥ 3 Points = Change from W3 to W4 in PCL ≥ 10, GERS, and BMI, Adjusting for Age Group at 9/11, Gender, Race/Ethnicity, Education, Exposure to the Disaster, Dust Cloud, Healthcare Accessibility, Smoking, History of Obstructive Sleep Apnea, and Asthma Control Level at W3, *n* = 1861					
		Bivariate Analysis		Adjusted Analysis	
		Change in ACT Score		Change in ACT Score	
	**Number**	**Worsened by ≥ 3 vs. No Change**	**Improved by ≥ 3 vs. No Change**	**Worsened by ≥ 3 vs. No Change**	**Improved by ≥ 3 vs. No Change**
** Measurement at W1 **					
**Age group on 11 September**					
18–34	645	ref	**ref**	ref	Ref
35–39	508	**1.34 (1.03, 1.75)**	0.80 (0.58, 1.11)	**1.45 (1.03, 2.04)**	0.63 (0.41, (0.98)
40–49	1026	**1.29 (1.02, 1.62)**	**0.73 (0.55, 0.95)**	**1.41 (1.04, 1.90)**	**0.59 (0.41, 0.86)**
50–	617	1.26 (0.98, 1.62)	**0.72 (0.53, 0.99)**	**1.61 (1.14, 2.26)**	**0.62 (0.40, 0.95)**
total	2796				
**Sex**					
male	1650	ref	ref	ref	Ref
female	1146	0.87 (0.73, 1.03)	**1.53 (1.24, 1.89)**	0.98 (0.75, 1.26)	**1.66 (1.18, 2.33)**
total	2796				
**Race/ethnicity**					
White, non-Hispanic	1880	ref	ref	ref	Ref
Black non-Hispanic, Hispanic, Asian, other	916	**1.23 (1.03, 1.46)**	1.24 (0.99, 1.55)	1.08 (0.84, 1.38)	0.73 (0.52, 1.01)
total	2796				
**Education**					
high school or less	672	**ref**	ref	0.94 (0.70, 1.26)	0.69 (0.44, 1.04)
some college	822	0.96 (0.72, 1.20)	1.11 (0.83, 1.49)	ref	Ref
college and higher	1287	**0.62 (0.50, 0.76)**	0.83 (0.64, 1.09)	**0.65 (0.51, 0.84)**	0.95 (0.68, 1.34)
total	2781				
**9/11 exposure category**					
RR worker	1623	ref	ref	ref	**Ref**
others	1173	**0.78 (0.66, 0.93)**	1.12 (0.90, 1.39)	**0.71 (0.55, 0.93)**	1.07 (0.75, 1.51)
total	2796				
**Dust cloud exposure**					
light	1123	ref	ref	ref	**Ref**
heavy	1667	1.11 (0.94, 1.32)	1.19 (0.96, 1.49)	1.15 (0.92, 1.44)	1.00 (0.74, 1.36)
	2790				
** Measurement at W3 **					
**Poor health care accessibility**					
no	2176	ref	ref	ref	**Ref**
yes	594	1.13 (0.91, 1.39)	**1.59 (1.24, 2.03)**	1.16 (0.871.53)	1.25 (0.89, 1.76)
total	2770				
**Smoking**					
never smoker	1378	ref	ref	ref	**Ref**
current smoker	258	1.34 (0.995, 1.80)	1.34 (0.92, 1.94)	0.85 (0.67, 1.08)	0.89 (0.65, 1.21
former smoker	972	0.93 (0.77, 1.13)	0.98 (0.77, 1.26)	1.18 (0.80, 1.73)	1.28 (0.78, 2.10)
total	2608				
**History of Obstructive sleep apnea**					
no	1856	ref	ref	ref	**Ref**
yes	805	**1.27 (1.05, 1.52)**	1.07 (0.84, 1.36)	**1.32 (1.03, 1.70)**	0.86 (0.62, 1.21)
total	2661				
**Asthma control test level at W3**					
controlled	1440	ref	ref	**ref**	Ref
poorly controlled	652	**1.74 (1.42, 2.14)**	**5.24 (3.90, 7.05)**	**1.55 (1.19, 2.03)**	**7.22 (4.91, 10.63)**
very poorly controlled	704	**0.76 (0.61, 0.95)**	**5.38 (4.08, 7.08)**	**0.52 (0.38, 0.71)**	**7.93 (5.37, 11.72)**
total	2796				
** Change from W3 to W4 **					
**Income level**					
no change	1383	ref	ref		
worsened	320	1.11 (0.85, 1.45)	0.85 (0.58, 1.24)		
improved	604	0.96 (0.77, 1.19)	1.09 (0.83, 1.43)		
	2307				
**GERS frequency change category**					
Remained the same	1950	ref	ref	ref	ref
Increased to ≥ twice a week	323	**1.60 (1.24, 2.07)**	1.02 (0.70, 1.46)	**1.70 (1.25, 2.31)**	0.86 (0.54, 1.36)
Decreased to < twice a week	198	0.91 (0.63, 1.30)	**1.58 (1.09, 2.30)**	0.84 (0,55, 1.29)	1.21 (0.77, 1.95)
total	2471				
**BMI change from W3 to W4**					
5% < change < -5%	1406	ref	ref	ref	Ref
increased by ≥ 5%	591	1.21 (0.98, 1.50)	1.03 (0.77, 1.37)	1.28 (0.98, 1.66)	0.95 (0.66, 1.36)
decreased by ≥ 5%	422	0.96 (0.75, 1.23)	1.15 (0.85, 1.56)	1.02 (0.76, 1.38)	1.19 (0.82, 1.73)
total	2419				
**Social support change from W3 to W4**					
no change	1463	ref	ref	ref	ref
worsened by ≥ 1 level	390	1.25 (0.98, 1.60)	1.11 (0.80, 1.54)	1.24 (0.93, 1.66)	0.85 (0.56, 1.27)
improved by ≥ 1 level	498	0.95 (0.75, 1.20)	1.13 (0.85, 1.51)	1.04 (0.79, 1.38)	1.08 (0.76, 1.54)
total	2351				
**PCL-17 score change from W3 to W4**					
within 10	1600	ref	ref	ref	ref
worsened by ≥ 10	230	**1.85 (1.37, 2.49)**	0.97 (0.61, 1.54)	**1.77 (1.26, 2.50)**	0.64 (0.37, 1.11)
improved by ≥ 10	426	1.08 (0.84, 1.39)	**2.00 (1.52, 2.65)**	1.00 (0.75, 1.35)	**1.42 (1.01, 1.99)**
total	2256				
Multivariable model: Change in ACT score of ≥ 3 points = Change from W3 to W4 in **PHQ ≥ 5**, GERS, and BMI, adjusting for age group at 9/11, gender, race/ethnicity, education, exposure to the disaster, dust cloud, healthcare accessibility, smoking, history of obstructive sleep apnea, and asthma control level at W3, *n* = 1842					
**PHQ-8 score change from W3 to W4**					
within 5	1693	ref	ref	**ref**	ref
worsened by ≥ 5	253	**1.53 (1.15, 2.03)**	0.93 (0.61, 1.42)	1.34 (0.96, 1.86)	0.68 (0.41, 1.12)
improved by ≥ 5	342	0.92 (0.70, 1.21)	**1.69 (1.25, 2.29)**	0.96 (0.70, 1.33)	1.23 (0.85, 1.78)
total	2288				

bolded text indicates *p* < 0.05.

**Table 5 ijerph-19-07795-t005:** Multivariable multinomial logistic regression, change in quality of life of 10 days poor health versus change in ACT score, W3 to W4.

**Model: Change in Days ≥10 of Poor Physical or Mental Health = Change in ACT Score (≥3), Change in PCL-17 (≥10), Change in BMI (≥5 Percent), Change in GERS Category; ACT Score, Obstructive Sleep Apnea, and Smoking at W3, and Sex, Age, Race/Ethnicity, and Education**				
**Days of good physical health**				
*n* = 1914	worsened by ≥10	improved by ≥10		
ACT worsened W3 to W4 ≥ 3	**1.97 (1.48, 2.62) ***	0.95 (0.66, 1.33)		
ACT improved W3 to W4 ≥ 3	0.80 (0.51, 1.26)	1.38 (0.96, 1.98)		
**Days of good mental health**			Removing PCL-17, *n* = 2083	
*n* = 1903	worsened by ≥10	improved by ≥10	worsened by ≥10	improved by ≥10
ACT worsened W3 to W4 ≥ 3	1.23 (0.89, 1.71)	0.92 (0.67, 1.26)	**1.44 (1.07, 1.96) ***	0.88 (0.66, 1.17)
ACT improved W3 to W4 ≥ 3	1.00 (0.63, 1.59)	0.98 (0.67, 1.43)	0.86 (0.56, 1.33)	1.05 (0.72, 1.49)
**Analysis Using 14-day Cutoff (Sensitivity Analysis to Demonstrate That Effects Are Similar to ≥ 10 Day Change)**				
**Model: Change to ≥14 Days of Poor Physical or Mental Health = Change in ACT Score (≥3), Change in PCL-17 (≥10), Change in BMI (≥5 percent), Change in GERS Category; ACT Score, Obstructive Sleep Apnea, and Smoking at W3, and Sex, Age, Race/Ethnicity, and Education**				
**Days of good physical health**				
*n* = 1914	worsened to ≥14	improved to <14		
ACT worsened W3 to W4 ≥ 3	**1.91 (1.37, 2.67)***	0.94 (0.62, 1.43)		
ACT improved W3 to W4 ≥ 3	0.79 (0.46, 1.37)	1.47 (0.94, 2.29)		
**Days of good mental health**			Removing PCL-17 *n* = 2083	
*n* = 1903	worsened to ≥14	improved to <14	worsened to ≥14	improved to <14
ACT worsened W3 to W4 ≥ 3	1.18 (0.78, 1.78)	1.00 (0.67, 1.48)	1.44 (0.98, 2.10)	0.84 (0.58, 1.21)
ACT improved W3 to W4 ≥ 3	1.10 (0.62, 1.95)	1.26 (0.81, 1.98)	0.90 (0.53, 1.55)	1.31 (0.87, 1.96)

* bolded text indicates confidence interval does not include 1.00.

## Data Availability

Unidentified data sets may be accessed from the New York City World Trade Center Health Registry (https://www1.nyc.gov/site/911health/researchers/health-data-tools.page, accessed on 10 June 2022). Researchers with further data may contain the WTCHR directly.

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
