# Peer review of "Change in Asthma Is Associated with Change in PTSD in World Trade Center Health Registrants, 2011 to 2016"

_ijerph, 2022, doi:10.3390/ijerph19137795_

Round 1

Reviewer 1 Report

The presented article describes research on the answer to the question whether change in Asthma is associated with change in PTSD in World 2 Trade Center Health Registrants. A huge advantage of the presented research is a very large group of patients exposed to negative mental experiences related to the tragic attack on the WTC.

The tests could be described in more detail in a separate section in the Materials and Methods section. The adopted values of changes in the assessed parameters (adopted for the logistic regression models) and the method of statistical analysis are correct. The tables contain numerous results of the analyzes.

The analysis of the results allowed for a positive confirmation of the thesis contained in the title of the article.

I have one comment to the presented research model. The latest data comes from the 4th wave, i.e. from 2016. It is quite a distant perspective considering the COVID-19 pandemic, which could have been of great importance for the issue under study. This seems to be a significant limitation of study.

Author Response

Reviewer

The tests could be described in more detail in a separate section in the Materials and Methods section.

Response

Several changes were made to the Methods.  A table was added to describe in detail how the dependent variable (ACT change) and change of risk factor variables (income level, smoking, GERS frequency, BMI relative change, social support, PTSD, PHQ score change, physical quality of life and mental quality of life) were calculated.

The variable for poor health care accessibility was defined as absence of health insurance, absence of a personal health care provider, or not receiving needed physical health care in the last 12 months.

Reviewer 

I have one comment to the presented research model. The latest data comes from the 4th wave, i.e. from 2016. It is quite a distant perspective considering the COVID-19 pandemic, which could have been of great importance for the issue under study. This seems to be a significant limitation of study.

Response

This study antedated the COVID-19 pandemic, but it may have an ongoing effect on WTCHR enrollees with PTSD and asthma, and their quality of life.  Subsequent WTCHR surveys will provide an opportunity to investigate the relationship between these health issues and COVID-19 in detail.

In addition, the text and tables have been reviewed and edited for clearly.

Thank you for your comments and suggestions.

Reviewer 2 Report

Comments:

This study tested whether clinical change in mental problems was associated with difference in asthma control, and whether change in asthma control varied with change in quality of life. For this purpose, data from the WTC Health Registry were used for statistical analysis. Although some limitations are due to the type of this study, the topic is interesting and the authors found that change in mental health measures tracked with change in asthma control, which correlated with change in quality of life.

The findings of this study may contribute to understand asthma risk factors and pathogeny, and to develop new strategies to improve asthma control.

Below are some comments:

Title/ Abstract

- PTSD is an abbreviation that was defined in part Introduction (Page 2: line 53-54)

Introduction:

- The authors reported data from the literature about asthma and its control in WTC disaster- exposed populations. They exposed the fact that effect of change in level of asthma control on health-related quality of life has not been reported as well as the effect of change in some risk factors on the level of asthma control.

Methods:

- The study population and the collected data sources in this study were appropriately reported apart from a few remarks:

- Page 2; Line 89: n = 10,583 participants (reported asthma in waves 1, 2 and 3) should be added in the text as it was made in figure 1.

- Page 3; Figure 1: “n = 5,379 participants who completed the asthma symptoms and control module in wave 3”:  This should be added in the second box of the figure.

- Page 3; Page 4: Line 111 - 146: If possible, it would be desirable to resume in a table the risk factors variables as well as their significant changes from W3 to W4.

Results:

- Results are presented in the text and in tables accordingly to the Protocol and statistical analysis described in Materials and Methods.

-  Page 7; Line 205: The title of table 1.b should be clarified.

- Page 7; Table1.b: “Distribution by ACT change from W3 to W4” was not mentioned in the upper part of table 1.b.

Discussion

The main results of this study were firstly reported. These results showed that clinically significant change over time in ACT score tracked clinically significant change in PTSD score. This finding was also evident with the bivariate and multivariable logistic regression.

Comparisons of the particularities and the findings of this study with regard to those of other studies were made and finally, the authors discussed some limitations of this study.

Author Response

Reviewer

PTSD is an abbreviation that was defined in part Introduction (Page 2: line 53-54)

Response

Posttraumatic stress disorder checklist (PCL-17).

Reviewer

  • Page 2; Line 89: n = 10,583 participants (reported asthma in waves 1, 2 and 3) should be added in the text as it was made in figure 1.

Response

There were 10,583 enrollees who reported that they had been diagnosed with asthma between 9/11/2001 through 2012 on the Wave 1, 2 or 3 surveys. 

Reviewer

  • Page 3; Figure 1: “n = 5,379 participants who completed the asthma symptoms and control module in wave 3”:  This should be added in the second box of the figure.

Response

Box revised.

Reviewer

  • Page 3; Page 4: Line 111 - 146: If possible, it would be desirable to resume in a table the risk factors variables as well as their significant changes from W3 to W4.

Response

Table was added as table 1.

Also variable "health care accessibility" was defined as W3 survey absence of health insurance, absence of a personal health care provider, or not receiving needed physical health care in the past 12 months.

Table 1. Change from Wave 3 to Wave 4

Variable

Significant change

Reference

ACT

Increase of 3 or more points or decrease of 3 or more points

Remain within 2 points

Income level

Increase or decrease of one or more levels

No change

Smoking

Started smoking or quit smoking

No change

GERS frequency in last 12 months

Increase to two or more times a week or decrease to less than two times a week

Not crossing the ≥ twice a week frequency threshold

BMI relative change

Increase by at 5% or more or decrease by 5% or more

Remain within 4.99%

Social support

Increase by 2 or more levels or decrease by 2 or more levels

Remain within 1 level

PTSD-17

Increase by 10 points or more or decrease by 10 points of more

Remain within 9 points

PHQ-8

Increase by 5 points or more or decrease by 5 points or more

Remain within 4 points

Physical health QoL

Increase of 10 days or more or decrease of 10 days or more

Remain within 9 days

Increase or decrease of days to cross the 14 days/month threshold

not crossing the 14 days/ month threshold

Mental health QoL

Increase of 10 days or more or decrease of 10 days or more

Remain within 9 days

Increase or decrease of days to cross the 14 days/month threshold

not crossing the 14 days/ month threshold

Reviwer

Results:

  •  Page 7; Line 205: The title of table 1.b should be clarified.

Response

Table 1.b is now table 2.b. Factors measured as the difference between W3 and W4.

Association between change in risk factor and change in ACT score.

able 1.b is now table 2.b-

Reviewer

Page 7; Table1.b: “Distribution by ACT change from W3 to W4” was not mentioned in the upper part of table 1.b.

Response

Added to table 2.b.
